# Efficacy of Prostate Biopsies via Transperineal and Transrectal Routes for Significant Prostate Cancer Detection: A Multicenter Paired–Matched Study

**DOI:** 10.3390/diagnostics15030288

**Published:** 2025-01-26

**Authors:** Nahuel Paesano, Natàlia Picola, Jesús Muñoz-Rodriguez, Xavier Ruiz-Plazas, Marta V. Muñoz-Rivero, Ana Celma, Gemma García-de Manuel, Berta Miró, Pol Servian, José M. Abascal, Enrique Trilla, Juan Morote

**Affiliations:** 1Clínica Creu Blanca, 08018 Barcelona, Spain; 2Department of Surgery, Universitat Autònoma de Barcelona, 08193 Bellaterra, Spain; enrique.trilla@vallhebron.cat (E.T.); juan.morote@uab.cat (J.M.); 3Department of Urology, Hospital Universitari de Bellvitge, 08907 Hospitalet de Llobregat, Spain; 4Department of Urology, Hospital Universitari Parc Tauli, 08208 Sabadell, Spain; jmunoz@tauli.cat; 5Department of Urology, Hospital Universitari Joan XXIII, 43005 Tarragona, Spain; xruiz.tarragona.ics@gencat.cat; 6Department of Urology, Hospital Universitari Arnau de Vilanova, 25198 Lleida, Spain; mvmunoz.lleida.ics@gencat.cat; 7Department of Urology, Hospital Univeritari Vall d’Hebron, 08035 Barcelona, Spain; ana.celma@vallhebron.cat; 8Department of Urology, Hospital Universitari Josep Trueta, 17007 Girona, Spain; gemmagarcia.girona.ics@gencat.cat; 9Statistic Unit, Vall d’Hebron Research Institute, 08035 Barcelona, Spain; berta.miro@vhir.org; 10Department of Urology, Hospital Univeritari Germans Trias i Pujol, 08916 Badalona, Spain; pservian.germanstrias@gencat.cat; 11Department of Urology, Parc de Salut Mar, 08003 Barcelona, Spain; jabascal@psmar.cat; 12Department of Medicine and Health Sciences, Universitat Pompeu Fabra, 08002 Barcelona, Spain

**Keywords:** prostate biopsy, magnetic resonance imaging, target biopsy, transperineal, transrectal, prostate cancer

## Abstract

**Background**: A transperineal approach to prostate biopsy is now recommended to reduce the risk of infectious complications associated with the transrectal route. Our aim is to compare the efficacy of transrectal- and transperineal-guided biopsies involving the magnetic resonance imaging (MRI) of index lesions in detecting significant prostate cancer (sPCas), and to evaluate the role of systematic biopsies. **Methods**: In a prospective and multicenter trial conducted in an opportunistic early detection program for sPCa in Catalonia (Spain), between 2021 and 2023, 4029 men suspected of having PCa underwent multiparametric MRI followed by guided and systematic biopsies. From this cohort, we retrospectively selected 1376 men with reports of the size and localization of their index lesions. A matched group of 325 pairs of men subjected to transrectal and transperineal biopsy were chosen to account for confounding variables. We compared sPCa detection rates determined via index lesions and systematic biopsies, as well as by lesion localization. **Results**: Transperineal and transrectal biopsies detected sPCa in 49.5% vs. 40.6% overall (*p* = 0.027), 44.6% vs. 30.8% from index lesions (*p* = 0.001), and 24.3% vs. 35.1% from systematic biopsies (*p* = 0.003). SPCa detection rates were higher in transperineal biopsies across all index lesion localizations, with significant increases in the anterior zone (47.8% vs. 20.8% at the mid-base, *p* = 0.039, and 52.9% vs. 24.2% at the apex, *p* = 0.024) and central zone (33.3% vs. 5.9%, *p* = 0.003). With regards to SPCa detected only in systematic biopsies, 10.5% of cases were detected in transrectal biopsies and 4.9% of cases were detected in transperineal biopsies (*p* = 0.012). **Conclusions**: Targeted biopsies conducted via the transperineal route showed higher sPCa detection rates than transrectal biopsies, particularly for anterior and apical lesions, with systematic biopsies showing reduced utility.

## 1. Introduction

Prostate cancer (PCa) is the second leading cause of mortality among men [1]. The introduction of multiparametric magnetic resonance imaging (MRI) has significantly improved early PCa detection, especially significant PCa (sPCa), thereby reducing morbidity and mortality through timely intervention [2]. Confirming the diagnosis of PCa typically involves a prostate biopsy, with techniques such as MRI–ultrasound (MRI-US) fusion-guided biopsy being employed to reduce false negatives [3]. However, the widespread global use of the transrectal approach presents considerable risks, including severe complications such as prostatitis, sepsis, and rectal bleeding [4]. In this scenario, Transperineal Prostate Biopsy (TPB) has emerged as an alternative to overcome some of the limitations of Transrectal Prostate Biopsy (TRB) that may increase the safety profile of the prostate biopsy procedure [5]. Concerns regarding infection risks and the prevalence of multidrug-resistant bacterial strains have led the European Association of Urology (EAU) to recommend a shift towards the transperineal route [6]. Reports indicate sepsis rates as low as zero with the transperineal approach [7], leading international groups to advocate for discontinuing the use of TRB as soon as possible [8]. However, a randomized trial conducted in the U.S., involving men undergoing either transrectal or transperineal prostate biopsy under local anesthesia, found complication rates of 2.6% and 2.7%, respectively. Importantly, no participants developed sepsis in either group [9]. The transperineal approach offers benefits, particularly in terms of anterior and apical sampling, whereas the far field suffers from degraded TRB resolution, making visualization challenging [10]. A study analyzing a modified transrectal-guided sextant biopsy, including anteriorly directed cores at the prostate apex, revealed the identification of 17% of tumors in the anterior and apical zones [11]. This critical consideration is currently being evaluated in the ongoing TRANSLATE trial, which randomly selects men suspected of having PCa to undergo MRI-targeted and systematic TRUS-guided biopsy via transrectal or transperineal routes, with the endpoint of diagnosing sPCa [12]. The targeted biopsy of index lesions, areas of the highest oncological suspicion, has proven to be highly effective, being capable of identifying more than 95% of sPCas. This highlights the importance of focusing evaluations on these critical areas [13]. In the era of MRI-targeted biopsy, controversies remain regarding the effectiveness of TPB and TRB. However, only a few studies have assessed the rates of detection of PCa and sPCa between these two approaches.

We hypothesize that the transperineal route for prostate biopsies is more effective than the transrectal route in targeting index lesions for sPCa detection in men suspected of having PCa. Our first objective is to analyze the efficacy of guided biopsies of the index lesion for detecting sPCa in a matched–paired group to avoid the influence of confounding variables. Secondly, we aim to compare the additional role of systematic biopsies in detecting sPCa according to the biopsy route. Finally, we seek to determine if specific localizations of index lesions benefit from either the transrectal or transperineal biopsy route.

## 2. Materials and Methods

### 2.1. Design, Setting, and Participants

This is a retrospective analysis of a prospective, multicenter trial conducted as part of the sPCa opportunistic screening program in Catalonia (Spain), between 2021 and 2023, in ten participating centers. The study involved 4029 men with serum prostate-specific antigen (PSA) levels higher than 3.0 ng/mL and/or a suspicious digital rectal examination (DRE), who underwent multiparametric magnetic resonance imaging (mpMRI), reported using the Prostate Imaging Reporting and Data System v. 2.1, followed by two-to-four-core-guided biopsies and/or twelve-core systematic biopsies. Exclusion criteria were applied in cases where data were unavailable regarding pathology in the index lesion and systematic biopsies, age (years), serum PSA levels (ng/mL), DRE (suspicious vs. normal), type of biopsy (initial vs. repeated), family history of PCa (no vs. yes), prostate volume (mL), Tesla scanner strength (1.5 vs. 3.0), PI-RADS (2–5), size of the index lesion (mm), index lesion location (posterior vs. anterior, and mid-base vs. apex), and type of fusion (cognitive vs. software-based). From this cohort, 1376 men were selected after index lesion size and localization were reported as well after the pathology of index lesions and that of systematic biopsies were separately determined. A 1:1 matched group of 325 pairs of men subjected to transrectal and transperineal biopsy was finally selected to account for confounding variables. The patient selection strategy is represented in Figure 1. The study received approval from the review board of the Vall d’Hebron Hospital coordinator center (PRAG-02/2021). All study participants signed written consent forms.

### 2.2. PCa Diagnostic Approach

Men suspected of having PCa were initially selected based on the recommendation of urologists at the primary health setting after tests showed serum prostate-specific antigen (PSA) levels higher than 3.0 ng/mL and/or a suspicious DRE. The men were then referred to the nearest participant centers. The MRI systems had magnetic field strengths of 1.5 Tesla in four centers and 3.0 Tesla in six centers, where an expert radiologist reported findings using PI-RADS v2.1 [14]. For all prostate biopsies, MRI-TRUS image fusion was performed, utilizing cognitive techniques and software techniques in five centers. Lesions with the highest PI-RADS scores and of the largest sizes were defined as MRI index lesions [13]. Biopsies were carried out transrectally in four centers and transperineally in six centers. Each suspected lesion underwent two-to-four-targeted cores according to their size, in addition to a twelve-core systematic biopsy using the classic sextant method per lobule, avoiding suspicious lesions in all men. Experienced operators conducted the biopsies in each center, and experienced uropathologists analyzed the biopsy materials, using the International Society of Urologic Pathology grade groups (ISUP-GGs). Insignificant PCa (iPCa) was defined by an ISUP-GG of 1, while sPCa was considered when the ISUP-GG was 2 or higher [15].

### 2.3. Variables in the Study

Age (years), serum PSA (ng/mL), digital rectal exam (DRE) characteristics (normal vs. suspicious), type of prostate biopsy (initial vs. repeated), first degree of PCa family history (no vs. yes), MRI prostate volume (mL), PI-RADS score (2–5), Tesla (1.5 vs. 3.0), number of suspicious lesions (1–3), length of index lesion (mm), posterior–anterior localization of index lesion (peripheral, central, and anterior zone), cranio-caudal localization of index lesion (mid-base vs. apex), type of fusion TRUS-MRI images (cognitive vs. software), type of biopsy approach (transrectal vs. transperineal), PCa, sPCa, and iPCa detected in the targeted biopsy of the index lesion and in systematic biopsy were all outcome variables.

### 2.4. Statistical Analysis

Participating centers provided anonymized datasets which were harmonized for statistical analysis. The study complied with Standards of Reporting for MRI-Targeted Biopsy Studies (START) guidelines [16]. Means and their 95% confidence intervals (CIs), and medians and interquartile ranges (IQR: 25th–75th percentile) described quantitative variables, while numbers and percentages described qualitative ones. Pearson’s chi-square and Mann–Whitney U tests compared qualitative and quantitative variables, respectively. Binary logistic regression identified independent predictors for sPCa, accounting for the prostate biopsy route. A 1:1 matching group was selected based on the biopsy route to normalize the effects of the confounding variables: age, serum PSA, suspicious DRE, repeated biopsies, family history of PCa, prostate volume, 3 Tesla scanner, PI-RADS, size of index lesion, index lesion location, and software-based use. R package matching v4.10, a multivariate and propensity score matching software with automated balance optimization (R Foundation for Statistical Computing, Vienna, Austria) was used. Significant differences were at *p* < 0.05; *p*-values from 0.05 to 0.1 indicated trends. This analysis was conducted with the Statistical Package for the Social Sciences (version 29.0, IBM Corp., Armonk, NY, USA).

## 3. Results

### 3.1. Characteristics of Cohort Study

The characteristics of the cohort study are summarized in Table 1. The median age was 68 years (IQR: 62–73), and the median PSA was 7.2 ng/mL (IQR: 5.3–10.8). Abnormal DRE findings were present in 25% of cases, repeated biopsies in 27.4%, and a family history of PCa in 10.8%. The median MRI prostate volume was 52 mL (IQR: 38–73), and the PSA density was 0.14 ng/mL/mL (IQR: 0.09–0.13). A 3 Tesla MRI was performed in 46.4% of cases, with two suspicious lesions identified in 23.9% and three in 3.8%. The PI-RADS score for the index lesion was 2 in 15.1% of cases, 3 in 20.3%, 4 in 43.3%, and 5 in 21.7%. The median index lesion length was 11 mm (IQR: 0.7–15). The index lesion was in the peripheral zone in 64.8% of cases, the central/transitional zone in 14.2%, and the anterior zone in 21%. Regarding the cranio-caudal axis, the index lesion was located at the mid-base in 59.2% of cases and at the apex in 40.8%. Software TRUS-MRI fusion images for targeted biopsies were used in 55.7% of cases. The transperineal route was used in 59.8% of cases, and the transrectal route in 40.2%. Overall, sPCa was detected in 41.3% of targeted biopsies of the index lesion and in 29.7% of systematic biopsies, while iPCa was detected in 13.1% and 18.5% of cases, respectively.

### 3.2. Binary Logistic Regression for Searching Independent Predictive Variables of sPCa, Selection of a Matched Group to Avoid Confounders, and Characteristics of Paired Groups

Odds ratios (ORs) and 95% confidence intervals (CIs) of the potential independent predictive variables for sPCa are presented in Table 2. It was found that age, serum PSA, DRE, type of biopsy, prostate volume, Tesla, type of TRUS-MRI fusion image, and PI-RADS were independent predictive variables for sPCa. To avoid the influence of these confounders on sPCa detection and normalize the overall influence of the prostate biopsy route according to the localization of the index lesion, a 1:1 matched group of 325 pairs of participants (650 cases) was selected. Table 3 summarizes the comparison of all variables included in the matched study group. All characteristics of participants, MRI, and biopsy techniques were statistically similar with respect to the peripheral localization vs. other localization of index lesions and the outcome variables sPCa and iPCa. The index lesion was localized in the peripheral zone in 49.2% of men who underwent transrectal biopsy and 50.8% of those who underwent the transperineal route, *p* = 0.793. Central/transitional zone localization was observed in 21.2% and 10.2%, respectively, at *p* = 0.032, and fibromuscular anterior zone localization was observed in 13.2% and 22.8%, respectively, at *p* = 0.028. Regarding the cranio-caudal axis, 48% of index lesions occupied the mid-base in men who underwent TRB and 63.4% occupied the mid-base in those subjected to TPB; index lesions occupied the apex in 52% and 36.6%, respectively, at *p* < 0.001. The overall detection of PCa in guided biopsies of the index lesion and systematic biopsies conducted through the transrectal route was 58.5%, while it was 67.1% when the biopsy was conducted through the transperineal route, *p* = 0.028.

### 3.3. Overall Efficacy of Systematic Biopsies According to the Prostate Biopsy Route

The overall detection rate of PCa in systematic biopsies when the transrectal route was utilized was 52.9%, while it was 48% when the transperineal route was used, at *p* = 0.239. Regarding sPCa detection, these rates were 35.1% and 24.3%, respectively, at *p* = 0.003; for iPCa detection, the rates were 17.8% and 23.7%, respectively, at *p* = 0.082. We note that 10.5% of sPCas were only detected in systematic biopsies when the transrectal route was conducted, while 4.9% of them were detected when the transperineal route was used, *p* = 0.012 (Table 4).

### 3.4. Overall Efficacy of Guided Biopsies to the Index Lesion According to the Biopsy Route and Localizations

The overall detection rates of PCa in the index lesion for the transrectal and transperineal routes were 43.1% and 61.5%, respectively, at *p* < 0.001. SPCa detection rates were 30.8% and 44.6%, respectively, at *p* < 0.001, and iPCa detection rates were 12% and 16.9%, respectively, at *p* = 0.094, as shown in Table 3.

The rate of sPCa detection in the recodified localizations of index lesions regarding the posterior–anterior and cranio-caudal axes and the prostate biopsy route are summarized in Table 5. We noted non-significant differences between both biopsy routes when the index lesion was localized in the peripheral zone, either at the mid-base or apex. TPB detected more sPCas than TRB did in the central and mid-base zones, at 33.3% and 5.9%, respectively, and at *p* = 0.003. In the anterior and mid-base zones, sPCa detection rates were 47.8% and 20.8%, respectively, at *p* = 0.039. In the anterior and apical zones, these rates were 44.6% and 24.2%, respectively, at *p* = 0.024.

## 4. Discussion

Technological advancements in prostate magnetic resonance imaging (MRI) have significantly transformed the diagnostic landscape, leading to higher detection rates of clinically significant prostate cancer (sPCa) [17]. However, direct comparisons of the effectiveness of transperineal biopsy (TPB) and transrectal biopsy (TRB), particularly with MRI/US fusion techniques, remain limited. This study aims to address this gap through a comprehensive analysis of sPCa detection rates, the role of systematic and targeted biopsies, and the impact of lesion localization in TPB and TRB in the MRI era.

Recent studies, such as the PREVENT Randomized Trial, have found no significant differences in sPCa detection between the transperineal and transrectal routes, although transperineal biopsies avoid infectious complications [18]. Similarly, Hoeh et al. concluded that transitioning from transrectal to transperineal MRI-guided prostate biopsy did not compromise the rates of detection of clinically significant prostate cancer (51% vs. 52%) in a tertiary care setting [19]. Conversely, Rai et al. demonstrated higher sPCa detection rates with TPB in a meta-analysis [20]. Similarly, Zattoni et al. reported significantly higher detection rates for TPB, identifying it as an independent predictor of sPCa with a detection rate of 49.1% compared to 35.2% for TRB [21].

Koparal et al., in a multicenter study comparing sPCa detection rates in TPB and TRB, matched 276 men undergoing TPB with 508 patients undergoing TRB by age, DRE, PSA density, and PI-RADS score. The results showed that MRI-targeted TPB and 12-core systematic TPB were significantly superior to TRB, with detection rates of 27.5% vs. 19.5% and 24.6% vs. 16.3%, respectively [22]. Similarly, Diamand et al. recently published a multicenter European study reporting higher detection rates using MRI-guided biopsy in the transperineal approach [23].

Kaneko et al., in a single-center pair-matched study, reported sPCa detection rates of 56% vs. 49%, which were significantly higher for TPB compared to TRB. When combining systematic and targeted biopsies, similar detection rates were observed: 59% for TPB and 60% for TRB [24].

In our multicenter pair-matched study, we observed sPCa detection rates of 30.8% and 44.6% in index lesions under the transrectal and transperineal routes, respectively. For combined systematic and targeted biopsies, the detection rates were 40.6% for TRB and 49.5% for TPB, which showed a significant difference. TPB exhibited significantly higher sPCa detection rates than TRB, particularly for index lesion targeting. The results obtained in our series are consistent with some of the previously described articles [20,21,22,23,24].

Detecting sPCa requires both systematic and targeted biopsies due to its multifocal nature and the risk of missing lesions with MRI [25]. The 2017 PROMIS trial showed that systematic biopsies alone have 48% sensitivity for sPCa, but combining them with targeted biopsies increases sensitivity to 93% [26]. The PRECISION study found that targeted biopsies detected sPCa in 38% of cases, compared to 26% with systematic biopsies alone [27]. The 2018 MRI-First study reported similar sensitivities for both methods [28]. The 2019 PAIREDCAP trial confirmed the necessity of using both biopsy types, as each method alone fails to detect all sPCa cases [29]. The GÖTEBORG-2 trial indicated that using transrectal MRI-targeted biopsies instead of systematic biopsies for patients with high PSA levels halved the risk of overdiagnosis but delayed the identification of some intermediate-risk tumors [30].

Few studies have compared the effectiveness of systematic biopsies between the transrectal and transperineal approaches in prostate fusion biopsies. In our analysis, systematic biopsies performed via the transrectal route demonstrated a higher detection rate of clinically significant prostate cancer (10.5%) compared to the transperineal route (4.9%). This finding, rarely explored in the literature, contrasts with studies such as those by Koparal et al. [22] and Zattoni et al. [21], which reported advantages of systematic biopsies via the transperineal route, particularly in larger prostates. These discrepancies may be attributed to methodological differences, population characteristics, or predominant prostate volume. In our study, prostates smaller than 30 cc. were associated with a significantly higher detection rates via the transrectal route, while the average prostate volume in both groups was approximately 60 cc., which could have influenced the detection rates observed. 

Our analysis also revealed that lesion location influences diagnostic outcomes. Apical and anterior prostate lesions were more effectively sampled in TPB due to better access, while TRB showed superior sampling of posterior lesions due to probe proximity and needle trajectory. No significant differences were noted in sPCa detection between the biopsy approaches for peripheral zone lesions. However, TPB was more effective in detecting sPCa in central and mid-base zones compared to TRB, with detection rates of 33.3% and 5.9%, respectively. Additionally, in the anterior and mid-base zones, detection rates were 47.8% and 20.8%, respectively, which were significant. For anterior and apical zones, the rates of sPCa detection were higher in TPB, with a 44.6% detection rate compared to 24.2% in TRB. This differential impact based on lesion location was further evidenced in subgroup analyses, showing higher sPCa detection rates for anterior and apical lesions with TPB. Consistent with our findings, a meta-analysis published by Uleri et al. found the transperineal approach superior for detecting sPCa in anterior and apical prostate tumors and found that it was equivalent in effectiveness to the transrectal route for peripheral tumors. The former demonstrated better precision for smaller anterior tumors than the transrectal method [31].

While our study benefits from a multicenter design and large sample size, it is not without limitations. The retrospective nature of some data could have introduced inherent selection biases, which we attempted to minimize using a matched-cohort design. Differences in biopsy protocols across various institutions could have generated variability in PCa detection, representing a limitation in this study. Although all centers followed current practice guidelines and terminology, the lack of a centralized review causes heterogeneity in the interpretation of MRIs and in the analysis of biopsies due to the involvement of multiple physicians. However, all MRI analyses were conducted by specialized genitourinary radiologists using version 2.1 of PI-RADS, and biopsy specimens were analyzed by experienced uropathologists at all centers. Our matched study was insufficient regarding certain locations, as differences were observed in some of them, and it was performed for peripheral locations versus other locations. Another potential limitation of our study is that it did not incorporated the results of secondary biopsied targeted lesions.

Future research should focus on conducting large prospective and randomized trials to determine the optimal biopsy approach in different clinical scenarios. These studies should avoid the influence of confounders, with especial attention to the localization and size of targeted lesions.

## 5. Conclusions

Targeted biopsies of index lesions performed via the transperineal route demons-trated higher detection rates of sPCa compared to those performed via the transrectal route in our cohort study. This improvement in the efficacy of targeted biopsies using the transperineal route was particularly notable for lesions located in the anterior and apical zones. The efficacy of systematic biopsies decreased when the transperineal route was utilized, especially when detecting sPCa with only them.

## Figures and Tables

**Figure 1 diagnostics-15-00288-f001:**
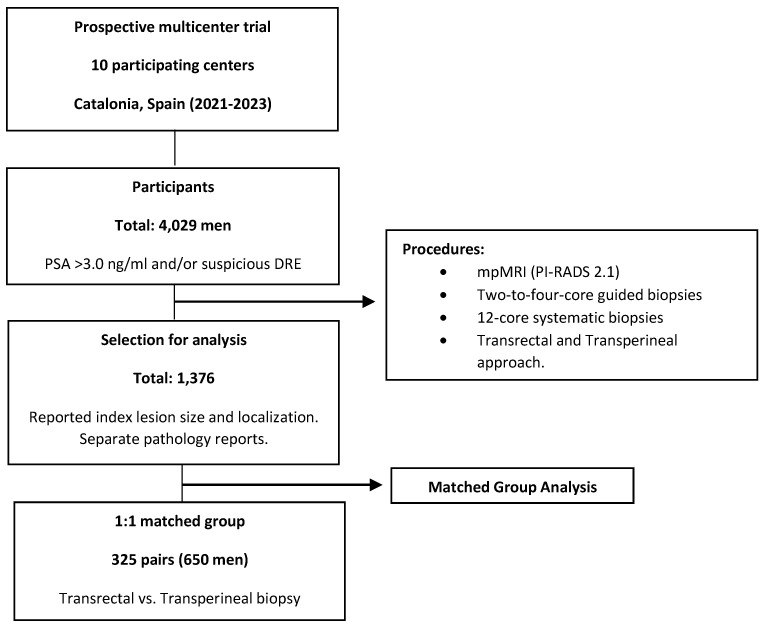
Flowchart of the patient selection strategy.

**Table 1 diagnostics-15-00288-t001:** Characteristics of the study population.

Characteristic	Measurement
Number of men	1376
Age, years, mean (95% CI)	67.5 (67.1–68.0)
Serum PSA, ng/mL, mean (95% CI)	9.7 (8.4–9.8)
Abnormal DRE, *n* (%)	344 (25.0)
Repeated prostate biopsy, *n* (%)	377 (27.4)
PCa family history, *n* (%)	149 (10.8)
Prostate volume, cc, mean (95% CI)	58.9 (57.3–60.5)
PSA density, ng/mL/cc, mean (95% CI))	0.14 (0.09–0.13)
3 Tesla mpMRI, *n* (%)	638 (46.4)
Suspicious lesions, *n* (%)	
1	1376 (100)
2	329 (23.9)
3	52 (3.8)
Index lesion PI-RADS score, *n* (%)	
2	355 (15.1)
3	449 (20.3)
4	960 (43.3)
5	468 (21.7)
Index lesion length, mm, mean (95% CI)	11.7 (10.6–13.2)
Postero-anterior index lesion localization, *n* (%)	
Peripheral zone	891 (64.8)
Central/transitional zone	196 (14.2)
Anterior zone	289 (21.0)
Craniocaudal index lesion localization, *n* (%)	
Mid-Base	815 (59.2)
Apex	561 (40.8)
Software image TRUS-MRI fusion biopsy, *n* (%)	767 (55.7)
Transperineal route, *n* (%)	823 (59.8)
Overall PCa detection, *n* (%)	867 (63.0)
sPCa	652 (47.4)
iPCa	215 (15.6)
PCa detected at index lesion biopsy, *n* (%)	749 (54.4)
sPCa	568 (41.3)
iPCa	180 (13.1)
PCa detected at systematic biopsies, *n* (%)	662 (48.1)
sPCa	408 (29.7)
iPCa	254 (18.5)

CI = confidence interval; PSA = prostate-specific antigen; DRE = digital rectal examination; PCa = prostate cancer; PI-RADS = Prostate Imaging Reporting and Data System; sPCa = significant PCa; iPCa = insignificant PCa.

**Table 2 diagnostics-15-00288-t002:** Logistic regression analysis for searching independent predictive variables of sPCa detection in index lesions, apart from the prostate biopsy route.

Predictive Variable	Odds Ratio (95% CI)	*p* Value
Age, Ref. one year	1.054 (1.035–1.073)	<0.001
Serum PSA, Ref. one ng/mL	1.035 (1.013–1.057)	0.001
DRE, Ref. normal	1.581 (1.152–2.170)	0.005
Type of biopsy, Ref. initial	0.627 (0.460–0.857)	0.003
PCa family history, Ref. no	1.472 (0.952–2.275)	0.082
Prostate volume, Ref. one mL	0.978 (0.972–0.983)	<0.001
Tesla, Ref. 1.5	1.296 (1.221–2.728)	<0.001
Number of suspicious lesions, Ref. 1	1.021 (0.943–1.167)	0.671
Length of index lesion, Ref. one mm	1.003 (0.981–1.026)	0.783
PI-RADS score of index lesion, Ref. 2	3.938 (3.201–4.845)	<0.001
Postero-anterior localization of index lesion, Ref. PZ	0.888 (0.777–1.015)	0.081
Cranio-caudal localization of index lesion, Ref, mid-base	0.907 (0.804–1.022)	0.110
Type of guided biopsy, Ref. cognitive	1.911 (1.439–2.539)	<0.001

CI = confidence interval; PSA = prostate-specific antigen; DRE = digital rectal examination; PCa = prostate cancer; PI-RADS = Prostate Imaging Reporting and Data System; PZ = peripheral zone.

**Table 3 diagnostics-15-00288-t003:** Compared characteristics of the matched selected groups based on the prostate biopsy route.

Characteristic	Transrectal	Transperineal	*p* Value
Number of men	325	325	-
Age, years, mean (95% CI)	67.8 (67.1–68.5)	67.4 (66.8–67.5)	0.896
Serum PSA, ng/mL, mean (95% CI)	9.6 (8.9–9.6)	9.3 (8.7–9.8)	0.988
Abnormal DRE, *n* (%)	65 (20.0)	68 (20.0)	1.000
Repeated prostate biopsy, *n* (%)	89 (27.4)	86 (26.5)	0.754
PCa family history, *n* (%)	39 (12.0)	43 (13.2)	0.838
Prostate volume, cc, mean (95% CI)	61.5 (58.9–59.3)	59.3 (55.3–59.2)	0.678
PSA density, ng/mL/cc, mean (95% CI)	0.17 (0.13–0.19)	0.16 (0.15–0.19)	0.980
3 Tesla mpMRI, *n* (%)	169 (52.0)	164 (50.5)	0.892
Suspicious lesions, *n* (%)			
1	375(100)	375 (100)	1.000
2	81 (21.6)	87 (23.2)	0.894
3	14 (3.7)	16 (4.3)	0.905
Index lesion PI-RADS score, *n* (%)			
2	3 (0.9)	3 (0.9)	1.000
3	72 (22.2)	72 (22.2)	1.000
4	193 (59.4)	193 (59.4)	1.000
5	57 (15.5)	57 (15.5)	1.000
Index lesion length, mm, mean (95% CI)	11.3 (10.8–11.9)	11.6 (11.1–12.1)	0.981
Posteroanterior index lesion localization, *n* (%)			
Peripheral zone	213 (49.2)	220 (50.8)	0.793
Central (central and transitional) zone	69 (21.2)	33 (10.2)	0.032
Anterior zone	43 (13.2)	72 (22.2)	0.028
Craniocaudal index lesion localization, *n* (%)			
Mid-base	156 (48.0)	206 (63.4)	<0.001
Apex	169 (52.0)	119 (36.6)	<0.001
Software image TRUS-MRI fusion biopsy, *n* (%)	164 (49.2)	158 (48.6)	0.897
Overall PCa detection, *n* (%)	190 (58.5)	218 (67.1)	0.028
sPCa	132 (40.6)	161 (49.5)	0.027
iPCa	58 (17.8)	57 (17.5)	0.997
PCa detected at index lesion biopsy, *n* (%)	140 (43.1)	200 (61.5)	<0.001
sPCa	100 (30.8)	145 (44.6)	<0.001
iPCa	39 (12.0)	55 (16.9)	0.094
PCa detected at systematic biopsies, *n* (%)	172 (52.9)	156 (48.0)	0.239
sPCa	114 (35.1)	79 (24.3)	0.003
iPCa	58 (17.8)	77 (23.7)	0.082

CI = confidence interval; PSA = prostate-specific antigen; DRE = digital rectal examination; PCa = prostate cancer; PI-RADS = Prostate Imaging Reporting and Data System; sPCa = significant PCa; iPCa = insignificant PCa.

**Table 4 diagnostics-15-00288-t004:** The effectiveness of systematic biopsies and targeted biopsies on the index lesion for detecting overall PCa, sPCa, and iPCa, based on the prostate biopsy route, in the matched selected groups.

Type of PCa	Systematic Biopsies			Targeted Biopsies		
	TR Route (*n* = 325)	TP Route (*n* = 325)	*p* Value	TR Route (*n* = 325)	TP Route (*n* = 325)	*p* Value
sPCa, *n* (%)	144 (35.1)	79 (24.3)	0.003	100 (30.8)	145 (44.6)	<0.001
iPCa, *n* (%)	58 (17.8)	77 (23.7)	0.082	39 (12.0)	55 (16.9)	0.094
Overall PCa, *n* (%)	172 (52.9)	156 (48.0)	0.239	140 (43.1)	200 (61.5)	<0.001

n = number; PCa = prostate cancer; sPCa = significant PCa; iPCa = insignificant PCa; TR = transrectal route; TP = transperineal route.

**Table 5 diagnostics-15-00288-t005:** Detection of sPCa on targeted biopsies of index lesions according to their localization and the prostate biopsy route.

Localization of Index Lesion	sPCa Detection	Transrectal	Transperineal	*p* Value
PZ-MB, *n* (%)	102/217 (47.0)	36/81 (44.4)	66/136 (48.5)	0.577
CZ-MB, *n* (%)	11/75 (14.7)	3/51 (5.9)	8/24 (33.3)	0.003
AZ-MB, *n* (%)	27/70 (38.6)	5/24 (20.8)	22/46(47.8)	0.039
PZ-AP, *n* (%)	79/221 (35.7)	48/136 (35.3)	31/85 (36.5)	0.886
AZ-AP, *n* (%)	26/67 (37.7)	8/33 (24.2)	18/34 (52.9)	0.024
All localizations, *n* (%)	245/650 (37.7)	100/325 (30.8)	145/325 (44.6)	>0.001

TR = transrectal; TP = transperineal; PZ = peripheral zone; CZ = central zone; AZ = anterior zone; MB = mid-base; AP = apical.

## Data Availability

All data obtained and analyzed for this clinical study are available from the corresponding author upon reasonable request.

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
