# Peer review of "Efficacy of Prostate Biopsies via Transperineal and Transrectal Routes for Significant Prostate Cancer Detection: A Multicenter Paired–Matched Study"

_diagnostics, 2025, doi:10.3390/diagnostics15030288_

Round 1
Reviewer 1 Report
Comments and Suggestions for Authors
Dear Authors,
Congratulations on your article and the effort dedicated to its creation. The study addresses a relevant comparison between transperineal and transrectal prostate biopsies in the context of detecting clinically significant prostate cancer. The methodology is robust, being prospective, multicentric, and employing matched groups, which minimizes biases.
The methodological design is rigorous, with a considerable sample size (1,376 men), and holds evident clinical relevance for improving prostate biopsy protocols.
The introduction is concise and well-written.
The study objectives are clearly and succinctly described.
Regarding the originality of the topic, it should be noted that the subject matter has already been addressed with solid results in other relevant articles in the literature. Nonetheless, the methodology employed is appropriate and strengthens the results obtained. In formal aspects, maybe review of the syntax and vocabulary employed is recommended.
However, there are significant aspects that must be reviewed for proper evaluation and a possible acceptance.
Materials and Methods:
- The study is described as a prospective, multicentric analysis, but it is unclear which study is being referred to. It is, presumably, a prospective cohort study, as indicated, but its nature appears retrospective, as noted in the limitations. This discrepancy must be clarified and corrected to ensure proper methodology.
- Regarding the methodology for targeted lesion biopsies, what criteria were used to determine whether 2 or 4 cores were taken per lesion? Similarly, what criteria guided the systematic biopsy? Was the classic sextants method used? Did all patients undergo both targeted and systematic biopsies?
- Did the systematic biopsy include the area of the targeted biopsy, or was it randomized to avoid it? Clarifying this detail is crucial to ensure that the systematic biopsy is not conflated with the targeted biopsy in that area.
- What criteria were used to decide whether a patient underwent a transperineal or transrectal biopsy? Was there randomization? Were there differences across centers?
- Inclusion and exclusion criteria were not mentioned. This is a crucial point that must be addressed and justified.
- Cognitive biopsy techniques are not equivalent to software-based techniques due to differences in sample collection methodology, which is highly operator-dependent. How was their validity ensured for the purposes of comparison?
- In the statistical analysis, central tendency measures such as the mean should be used whenever possible. If the median and interquartile range are employed, their use must be justified as they indicate deviation from the mean.
- To achieve proper randomization and matching of patients, a strict definition of the criteria used for 1:1 pairing is necessary. This is not addressed but should be included as part of the study methodology. Additionally, the rationale for excluding certain patients must be specified. Of the 1,376 selected patients, why were only 650 studied? What happened to the rest? A detailed explanation is required to avoid potential selection bias.
Results:
- It is noteworthy that 21% of index lesions are located in the anterior fibrous zone, while only 14.2% are in the transitional zone. This finding should be highlighted and, if possible, justified.
- Regarding the percentages of sPCa detected in targeted versus systematic biopsies, clarify whether the systematic biopsy might have included samples from the targeted lesion, potentially leading to "contamination" in its determination (a common issue with cognitive biopsies).
- In Table 2, significant differences are shown in the use of 1.5T versus 3T MRI systems, with an OR of 1.296 favoring 3T. What relevance did this have for the 1:1 patient randomization? Additionally, biopsies performed with fusion software versus without showed an OR of 1.911. Does this imply that cognitive approaches are more effective in detecting significant tumors? This finding requires explanation.
- The patient selection for comparison is appropriate, as reflected in Table 3. However, it is unclear why 726 patients were excluded. Was it due to significant deviations or missing data? This could lead to substantial selection bias, which must be justified in detail.
- Section 3.4 and Table 5 highlight the importance of the transperineal approach for optimal diagnosis of prostate cancer in the anterior and transitional zones. This should be emphasized.
- It is interesting that systematic biopsies via the transrectal route detect 10.5% more significant tumors compared to the transperineal route (4.9%). What could be the reason for this? Greater ability to biopsy the base, the most frequent site, perhaps?
Discussion:
- The text addresses a complex scientific topic with numerous data points and cited studies. However, the structure feels somewhat overwhelming due to the large volume of information presented without clear separation between main and secondary ideas. Subheadings or section divisions are recommended to improve readability. Better synthesis of information is also advisable.
- From line 273 onward, the observation that systematic biopsies via the transrectal route detect 10.5% more significant tumors compared to the transperineal route (4.9%) is intriguing. What could explain this? Greater ability to biopsy the base, perhaps?
- This section could benefit from improved structure: Comparison of TPB vs. TRB, Importance of Combining Targeted and Systematic Biopsies, Impact of Lesion Location, Limitations and Future Directions.
- The limitations are highly relevant. You state that the study is retrospective, whereas in the methodology section, it is described as prospective. This discrepancy is critical, not only for the inconsistency but also for its implications for validity. If it is a prospective study approved by an Ethics Committee, there should be an established and rigorous biopsy protocol already indicated in the project. Indicating variability in biopsy models and protocols across study centers raises questions about the results, as this heterogeneity could invalidate the data provided. The phrase “The lack of centralized review” is particularly concerning in terms of generalizing results and could compromise the validity of the study.
- The conclusions are accurate but should explore possible reasons for the higher prevalence of significant tumors detected in systematic transrectal biopsies compared to the transperineal approach. This finding may not be generalizable and, therefore, should not form part of the conclusions, as it appears to be an isolated result of this study.
- Although the references are well-incorporated, there is redundancy in the comparison of similar studies. Consolidating common findings would reduce repetition.
Author Response
We appreciate comments and suggestions of the Reviewer, and the opportunity to make modifications in the manuscript which are highlighted in red.
Overall comment:
Congratulations on your article and the effort dedicated to its creation. The study addresses a relevant comparison between transperineal and transrectal prostate biopsies in the context of detecting clinically significant prostate cancer. The methodology is robust, being prospective, multicentric, and employing matched groups, which minimizes biases.
The methodological design is rigorous, with a considerable sample size (1,376 men), and holds evident clinical relevance for improving prostate biopsy protocols.
The introduction is concise and well-written.
The study objectives are clearly and succinctly described.
Regarding the originality of the topic, it should be noted that the subject matter has already been addressed with solid results in other relevant articles in the literature. Nonetheless, the methodology employed is appropriate and strengthens the results obtained. In formal aspects, maybe review of the syntax and vocabulary employed is recommended.
Response: Thank you.
However, there are significant aspects that must be reviewed for proper evaluation and a possible acceptance.
Materials and Methods:
- The study is described as a prospective, multicentric analysis, but it is unclear which study is being referred to. It is, presumably, a prospective cohort study, as indicated, but its nature appears retrospective, as noted in the limitations. This discrepancy must be clarified and corrected to ensure proper methodology.
Response: Thank you for your comment. This study was indeed a retrospective analysis conducted within the framework of a prospective study involving men suspected of having PCa who underwent mpMRI as well as both guided and systematic prostate biopsies. The nature of this study is clarified in abstract (lines 35-39) and the Materials and Methods section (lines 96-98).
- Regarding the methodology for targeted lesion biopsies, what criteria were used to determine whether 2 or 4 cores were taken per lesion? Similarly, what criteria guided the systematic biopsy? Was the classic sextants method used? Did all patients undergo both targeted and systematic biopsies?
Response: Thank you. Two to four cores were obtained in targeted biopsies based on lesion size. Twelve-core systematic biopsies were performed in all cases using the classic sextant method per lobule, avoiding suspicious lesions. This information is now specified in lines 120-123.
- Did the systematic biopsy include the area of the targeted biopsy, or was it randomized to avoid it? Clarifying this detail is crucial to ensure that the systematic biopsy is not conflated with the targeted biopsy in that area.
Response: Thank you. Systematic biopsies were randomized to avoid suspicious lesions. This is clarification is provided in lines 12-129.
- What criteria were used to decide whether a patient underwent a transperineal or transrectal biopsy? Was there randomization? Were there differences across centers?
Response: As specified in lines 126-127, the TR and TP routes were not randomized. Four centers exclusively utilized the TR route, while six participant centers employed the TP route.
- Inclusion and exclusion criteria were not mentioned. This is a crucial point that must be addressed and justified.
Response: Thank you. Men included in this analysis were those who underwent targeted and systematic biopsies. Exclusion criteria were applied in cases where data were unavailable regarding pathology at the index lesion and systematic biopsies, age (years), serum PSA levels (ng/mL), DRE (suspicious vs normal), type of biopsy (initial vs. repeated), family history of PCa (no vs. yes), prostate volume (mL), Tesla scanner strength (1.5 vs. 3.0), PI-RADS (2-5), size of the index lesion (mm), index lesion location (posterior vs. anterior, and mid-base vs. apex), and type of fusion (cognitive vs. software-based, (lines 102-108).
- Cognitive biopsy techniques are not equivalent to software-based techniques due to differences in sample collection methodology, which is highly operator-dependent. How was their validity ensured for the purposes of comparison?
Response: We appreciate this comment and agree with the Reviewer; however, differences between cognitive and software-based fusion biopsies have not been definitively established (Khoo, C.C et al. A Comparison of Prostate Cancer Detection between Visual Estimation, Cognitive Registration, and Image Fusion (Software Registration) Targeted Transperineal Prostate Biopsy. J Urol. 2021 Apr, 205, 1075-1081.
Nonetheless, the use of software-based biopsies was included as a variable in the selection of matched groups (line 144-149). The rate of software-based biopsies was comparable between the two paired groups analyzed (49.2 in TRB vs.48.6%, p =0.897, Table 3).
- In the statistical analysis, central tendency measures such as the mean should be used whenever possible. If the median and interquartile range are employed, their use must be justified as they indicate deviation from the mean.
Response: Thank you. Since biological variables are typically not normally distributed, reporting the median and IQR is often informative. However, we agree with the Reviewer´s suggestion. We have now included the mean and its 95% confidence interval for quantitative variables in the tables, while maintaining the reporting of medians and IQR within the manuscript. This has been clarified in lines 146-147.
- To achieve proper randomization and matching of patients, a strict definition of the criteria used for 1:1 pairing is necessary. This is not addressed but should be included as part of the study methodology. Additionally, the rationale for excluding certain patients must be specified. Of the 1,376 selected patients, why were only 650 studied? What happened to the rest? A detailed explanation is required to avoid potential selection bias.
Response: Thank you. The criteria used for 1:1 paring are now described in lines 152-154, resulting in 325 paired cases. No men were excluded after paired group selection. The remaining men were not paired.
Results:
- It is noteworthy that 21% of index lesions are located in the anterior fibrous zone, while only 14.2% are in the transitional zone. This finding should be highlighted and, if possible, justified.
Response: We agree with the Reviewer. However, the high distribution of index lesion in the anterior zone may be influenced by the presence of several peripheral lesions. We have replaced the “anterior fibrous zone” with anterior zone”, as this term is more appropriate.
- Regarding the percentages of sPCa detected in targeted versus systematic biopsies, clarify whether the systematic biopsy might have included samples from the targeted lesion, potentially leading to "contamination" in its determination (a common issue with cognitive biopsies).
Response: We agree with the Reviewer; however, the overall consensus among participating centers was to avoid systematic biopsies in suspicious lesions (lines 126-127)
- In Table 2, significant differences are shown in the use of 1.5T versus 3T MRI systems, with an OR of 1.296 favoring 3T. What relevance did this have for the 1:1 patient randomization? Additionally, biopsies performed with fusion software versus without showed an OR of 1.911. Does this imply that cognitive approaches are more effective in detecting significant tumors? This finding requires explanation.
Response: Thank you. Table 2 presents the logistic regression analysis performed to identify confounding variables for detecting sPCa in targeted biopsies of index lesions. This analysis was conducted to determine which variables should be considered when generating the 1:1 paired matched group from the 1.376 men initially selected. Men who underwent mpMRI using a 3Tesla scanner had an OR of 1.296 compared to those in who were scanned with a 1.5 T scanner. Additionally, men who underwent software-based fusion biopsy, compared with those using cognitive fusion biopsy, presented an OR of 1.911. It was an error to show “Ref. software” instead of “cognitive” which has now been corrected. We apologize for any inconvenience caused.
- The patient selection for comparison is appropriate, as reflected in Table 3. However, it is unclear why 726 patients were excluded. Software as it due to significant deviations or missing data? This could lead to substantial selection bias, which must be justified in detail.
Response: Table 3 analyzes the characteristics of the selected paired matched group. The suitability of the matched group was verified by the similarity of confounding variables identified in the previous logistic regression analysis. Due to the presence of several confounding variables, 726 men were excluded in order to form the 1:1 paired matched group. This was an essential step for analyzing the true influence of the biopsy route.
- Section 3.4 and Table 5 highlight the importance of the transperineal approach for optimal diagnosis of prostate cancer in the anterior and transitional zones. This should be emphasized.
Response: We agree with the Reviewer. This analysis is important, which is why it occupies a section in the Results (lines 226-239). This finding is highlighted in the Discussion section, specifically in lines 309–316.
- It is interesting that systematic biopsies via the transrectal route detect 10.5% more significant tumors compared to the transperineal route (4.9%). What could be the reason for this? Greater ability to biopsy the base, the most frequent site, perhaps?
Response: This is an interesting question. We believe that the different needle paths in TR and TP systematic biopsies result in lower detection of sPCa in the TP route, particularly in larger prostates, as the needle passes through the central zone. The TR route is ideal for biopsying the postero-lateral peripheral zone, whereas in the TP route, many cores are obtained from the central zone, where the incidence of sPCa is lower.
Following your observation, we analyzed sPCa detection in systematic biopsies according to the route and prostate volume in TR and TP routes. We observed a significantly higher detection rate of clinically significant prostate cancer (sPCa) in the TR route when prostate volumes were below 30 cc. However, from 31 cc and beyond, as prostate volume increases, the detection rate of clinically significant prostate cancer in systematic transrectal biopsies decreases, whereas it increases in systematic biopsies performed via the transperineal route. Given that we have a reasonably large prostate volume (means of approximately 60 cc in both groups), this could explain these findings.
Our results show that systematic biopsies performed via the transrectal route detected 10.5% more clinically significant tumors compared to the transperineal route (4.9%). This contrasts with the findings of Koparal et al., who reported that systematic transperineal biopsies have a higher detection rate of clinically significant prostate cancer, particularly in larger prostates. Similarly, Zattoni et al. observed that target biopsies combined with systematic biopsies via the transperineal route showed advantages in detecting tumors located in specific areas, such as the anterior zone and the apex.
The discrepancy between our results and those reported in these studies could be attributed to methodological differences, population characteristics, or predominant prostate volume. In our study, prostate volumes under 30 cc showed a significantly higher detection rate of sPCa in systematic biopsies via the transrectal route compared to the transperineal route. The mean prostate volume in both groups was approximately 60 cc, which could influence the detection rates observed and partially explain these differences. Comment places in lines 296-301.
Discussion:
- The text addresses a complex scientific topic with numerous data points and cited studies. However, the structure feels somewhat overwhelming due to the large volume of information presented without clear separation between main and secondary ideas. Subheadings or section divisions are recommended to improve readability. Better synthesis of information is also advisable.
Response: The discussion has been improved, and we have rewritten parts of the section to ensure better clarity and alignment with the suggested structure. We followed the reviewer recommendation to address the comparison between TPB and TRB, the importance of combining targeted and systematic biopsies, the impact of lesion location, as well as the limitations and future directions. We believe the revised text provides a more coherent flow and a clearer synthesis of the information.
- From line 273 onward, the observation that systematic biopsies via the transrectal route detect 10.5% more significant tumors compared to the transperineal route (4.9%) is intriguing. What
could explain this? Greater ability to biopsy the base, perhaps?
Response: As addressed in the previous response, our study found that systematic transrectal biopsies detect 10.5% more clinically significant tumors than transperineal biopsies, likely due to differences in needle trajectory and the prostate volumes analyzed. Comment places in lines 290-301.
- This section could benefit from improved structure: Comparison of TPB vs. TRB, Importance of Combining Targeted and Systematic Biopsies, Impact of Lesion Location, Limitations and Future Directions.
Response: Thank you for your suggestion. We have restructured the section accordingly to address the comparison of TPB vs. TRB, the importance of combining targeted and systematic biopsies, the impact of lesion location, as well as limitations and future directions
- The limitations are highly relevant. You state that the study is retrospective, whereas in the methodology section, it is described as prospective. This discrepancy is critical, not only for the inconsistency but also for its implications for validity. If it is a prospective study approved by an Ethics Committee, there should be an established and rigorous biopsy protocol already indicated in the project. Indicating variability in biopsy models and protocols across study centers raises questions about the results, as this heterogeneity could invalidate the data provided. The phrase “The lack of centralized review” is particularly concerning in terms of generalizing results and could compromise the validity of the study.
Response: Thank you very much for your comment. This study is a retrospective analysis nested within a prospective trial of men suspected of having prostate cancer who underwent mpMRI and guided systematic prostate biopsies in a multicentric study conducted across referral centers in Catalonia, Spain. The study design is clarified in the abstract (lines 35–39) and the Materials and Methods section (lines 96–98).
The study was approved by the Ethics Committee of Vall d'Hebron Hospital (Project Number: PRAG-02/2021), with approval granted on February 12, 2021. A standardized biopsy protocol was established at the outset of the trial, ensuring consistency across all participating centers. However, differences in biopsy protocols across various institutions could have generated variability in PCa detection, representing a limitation in this study (lines 319-321
- The conclusions are accurate but should explore possible reasons for the higher prevalence of significant tumors detected in systematic transrectal biopsies compared to the transperineal approach. This finding may not be generalizable and, therefore, should not form part of the conclusions, as it appears to be an isolated result of this study.
Response: Thank you very much. We have included these considerations throughout the discussion of the manuscript, trying to ensure that the conclusions are presented in a concise and clear manner, highlighting that they are in our study cohort (lines 336)
- Although the references are well-incorporated, there is redundancy in the comparison of similar studies. Consolidating common findings would reduce repetition.
Response: Thank you very much. In addition to the meta-analysis, we have included some individual articles from it, as they provided specific and valuable data. We have also added a new article to the bibliography following the suggestion of Reviewer 2.
Reviewer 2 Report
Comments and Suggestions for Authors
The authors aimed to compare the efficacy of transrectal and transperineal prostate-guided biopsies to magnetic resonance imaging (MRI) index lesions in detecting clinically significant prostate cancer (csPCa), and to evaluate the role of systematic biopsies. Despite the topic is still discussed in the most recent guidelines, this prospective and multicenter trial, conducted in Catalonia (Spain) between 2021 and 2023, involved 4,029 men suspected of having PCa who underwent multiparametric MRI followed by guided and systematic biopsies.
Briefly their results are confirmatory of the current knowledge.. Indeed they found that "Targeted biopsies via the transperineal route showed higher csPCa detection rates than transrectal biopsies, particularly for anterior and apical lesions, with systematic biopsies showing reduced utility."
The methodology is robust. However several improvements should be addressed.
First, PIRADS 2 if not abnormal at DRE should be excluded by the current analysis. A subgroup analysis should be performed to enrich the manuscript such as in PCA patients with familiarity, in PIRADS 3 lesions as well as anterior tumors.
Second, more than one subgroup should be completed according to PSA density and PIRADS consistently with new guidelines direction. It may be a point of strength of the current manuscript that as it stands now it is just largely confirmatory.
Third, new papers on PCA detection should be cited and discussed (PMID 36984626, 38893710 ,36165471, 39069444)
Author Response
We appreciate comments and suggestions of the Reviewer, and the opportunity to make modifications in the manuscript which are highlighted in red.
Overall comment:
The authors aimed to compare the efficacy of transrectal and transperineal prostate-guided biopsies to magnetic resonance imaging (MRI) index lesions in detecting clinically significant prostate cancer (csPCa), and to evaluate the role of systematic biopsies. Despite the topic is still discussed in the most recent guidelines, this prospective and multicenter trial, conducted in Catalonia (Spain) between 2021 and 2023, involved 4,029 men suspected of having PCa who underwent multiparametric MRI followed by guided and systematic biopsies.
Briefly their results are confirmatory of the current knowledge. Indeed, they found that "Targeted biopsies via the transperineal route showed higher csPCa detection rates than transrectal biopsies, particularly for anterior and apical lesions, with systematic biopsies showing reduced utility."
Response: Thank you.
The methodology is robust. However, several improvements should be addressed.
First, PIRADS 2 if not abnormal at DRE should be excluded by the current analysis. A subgroup analysis should be performed to enrich the manuscript such as in PCa patients with familiarity, in PIRADS 3 lesions as well as anterior tumors.
Response: Thank you. The inclusion criteria for this retrospective study were men suspected of having prostate cancer (PCa) who underwent targeted biopsy of suspicious lesions and systematic biopsy. Exclusion criteria included the unavailability of certain confounding variables, such as age (years), serum PSA levels (ng/mL), digital rectal examination (DRE; suspicious vs. normal), type of biopsy (initial vs. repeated), family history of PCa (no vs. yes), prostate volume (mL), Tesla scanner strength (1.5 vs. 3.0), PI-RADS (2–5), size of the index lesion (mm), index lesion location (posterior vs. anterior, and mid-base vs. apex), and type of fusion (cognitive vs. software-based). These data were essential for selecting the 1:1 paired matched group. Ultimately, we included three men in each group with PI-RADS 2 who underwent both targeted and systematic biopsy due to a suspicious DRE, elevated PSA density, or a family history of PCa.
We present the sPCa detection in guided biopsies of index lesions according to their location, including anterior lesions, in Section 3.4 (lines 220–233) and Table 5. Regarding PI-RADS scores, we observed that sPCa detection in the TP route is higher in PI-RADS 3 and 4, while it is comparable in PI-RADS 5 (data not shown). Concerning family history of PCa, we found that the TP route detected more sPCa in the subset with a family history than in those without (data not shown)
Second, more than one subgroup should be completed according to PSA density and PIRADS consistently with new guidelines direction. It may be a point of strength of the current manuscript that as it stands now it is just largely confirmatory.
Response: Thank you for this comment. PSA density (PSAD) was considered in this study only as a characteristic of the overall group and the paired matched group. The Reviewer suggests comparing the efficiency of the biopsy route for sPCa detection according to PSAD in men with PI-RADS 3.
We have now compared the efficacy of the TR and TP routes for guided biopsies of PI-RADS 3 index lesions based on a PSAD threshold of 0.15, in accordance with the new guideline recommendations. The TP route showed higher detection rate of sPCa, even in cases with PSAD 0.15 or lower, as well as in those with a PSAD above 0,15 (1.7% vs. 14% and 2.4% vs.32.3%, respectively, data not shown)
Third, new papers on PCA detection should be cited and discussed (PMID 36984626, 38893710 ,36165471, 39069444)
Response: Thank you very much for your suggestion. From the PMID reported by the Reviewer, The reference: Hoeh B, Wenzel M, Humke C, et al. "Transition from Transrectal to Transperineal MRI-Fusion Prostate Biopsy Does Not Compromise Detection Rates of Clinically Significant Prostate Cancer at a Tertiary Care Center." Diagnostics (Basel). 2024; 14:1184 has been incorporated in the discussion of the manuscript as reference number 19.
Round 2
Reviewer 1 Report
Comments and Suggestions for Authors
Thanks for the answers and corrections, everything is fine.
Reviewer 2 Report
Comments and Suggestions for Authors
No further comments are needed.